# Junction Temperature Optical Sensing Techniques for Power Switching Semiconductors: A Review

**DOI:** 10.3390/mi14081636

**Published:** 2023-08-19

**Authors:** Ridwanullahi Isa, Jawad Mirza, Salman Ghafoor, Mohammed Zahed Mustafa Khan, Khurram Karim Qureshi

**Affiliations:** 1Optical Communications and Sensors Laboratory (OCSL), Electrical Engineering Department, King Fahd University of Petroleum and Minerals, Dhahran 31261, Saudi Arabia; g202110270@kfupm.edu.sa; 2SEECS Photonics Research Group, Islamabad 44000, Pakistan; 3School of Electrical Engineering and Computer Science, National University of Sciences and Technology (NUST), Sector H-12, Islamabad 44000, Pakistan; 4Optoelectronics Research Laboratory, Electrical Engineering Department, King Fahd University of Petroleum and Minerals, Dhahran 31261, Saudi Arabia; 5Center for Communication Systems and Sensing, King Fahd University of Petroleum and Minerals, Dhahran 31261, Saudi Arabia

**Keywords:** junction temperature, electroluminescence, FBG, IGBT, SiC, MOSFET

## Abstract

Recent advancements in power electronic switches provide effective control and operational stability of power grid systems. Junction temperature is a crucial parameter of power-switching semiconductor devices, which needs monitoring to facilitate reliable operation and thermal control of power electronics circuits and ensure reliable performance. Over the years, various junction temperature measurement techniques have been developed, engaging both non-optical and optical-based methods, highlighting their advancements and challenges. This review focuses on several optical sensing-based junction temperature measuring techniques used for power-switching devices such as metal-oxide-semiconductor field-effect transistors (MOSFETs) and insulated-gate bipolar transistors (IGBTs). A comprehensive summary of recent developments in infrared camera (IRC), thermal sensitive optical parameter (TSOP), and fiber Bragg grating (FBG) temperature sensing techniques is provided, shedding light on their merits and challenges while providing a few possible future solutions. In addition, calibration methods and remedies for obtaining accurate measurements are discussed, thus providing better insight and directions for future research.

## 1. Introduction

Power switching semiconductors are indispensable elements in inverters and converters used in power grids/systems, automobiles, data centers, and renewable energy for reliability towards a more intelligent control system. While conventional switching devices applications are limited due to low switching speed and massive size, power switching semiconductors exhibit fast switching that can meet the load requirements and operating frequency of today’s technology [1,2]. Nowadays, over 1000 gigawatts of renewable energy incorporated into the power grids is controlled by power-switching semiconductors [3]. Additionally, power electronic converters and switches, which contain semiconductor devices, are utilized to regulate almost 60% of the supplied electrical energy consumed in industrialized countries [4]. Nevertheless, during power and thermal cycling, one of the most common failures is the wear-out caused by thermal stress on these power-switching semiconductor devices due to variations in their junction temperature [5,6,7,8]. Hence, real-time temperature sensing of circuits, including these devices, is of paramount importance. Recently, composite power switching devices such as IGBT- and Silicon Carbide (SiC)-based MOSFETs have gathered attention due to their improved performance characteristics (higher frequency operation, current handling capacity, and switching speed) than the conventional MOSFETs and bipolar junction transistors (BJT) [9]. 

Under normal operating conditions, the temperature of the semiconductor power switching devices rise above the ambient around the junction and other parts of the circuit. Furthermore, these devices are subjected to repetitive heat pulses due to high-frequency switching, which leads to the junction temperature variations as a result of the power losses [10]. This high junction temperature variations could be excessive and hamper the performance of the device due to overloading and abrupt changes in the input and output power of the circuit [3]. Although there is a relationship between the performance of an electronic component and its range of operating temperature, it has been well established in the literature that the susceptibility of an electronic assembly to fail increases exponentially with the junction temperature [11]. As such, this may lead to devastating effects of shutting down the power grid/system. Hence, several techniques have been proposed in the literature to measure the junction temperature of these critical power switching semiconductor devices. These techniques can be broadly classified as electrical-, physical-, and optical-based as illustrated in Figure 1.

The electrical-based technique uses electrical devices or electrical parameters for temperature measurement. Typical electrical devices include thermal-sensitive electrical devices (TSED) that employ additional electronic components such as resistors, diodes, and externally designed electrical circuits for measurement. Although this method provides excellent spatial resolution, it requires high costs and adds to the system’s complexity [10]. The temperature-sensitive electrical parameters (TSEP), such as the gate threshold voltage, saturation current, and short circuit current, are also suitable for online junction temperature sensing, but incur power loss to the system and thus are not suitable for measurement when the device is in operation. Another disadvantage of TSEP is that the device’s temperature distribution cannot be obtained since this measurement provides a point temperature value of the chip [12,13]. 

On the other hand, physical techniques include a thermistor and thermocouple (TC) that measure temperature differences, which are external to the system. Their techniques are simple to implement with excellent spatial resolution; however, the slow response in the measurement, especially in high-frequency circuits, remains the constraint for its deployment. In addition, this approach is practically difficult since the temperature measurement requires direct probe contact with the semiconductor device; thus, disassembling power circuits is unavoidable [14,15]. 

Recently, optical-based sensing (OBS) techniques have taken center stage as a viable non-invasive electromagnetic interference (EMI) immune junction temperature sensing technology, as highlighted in Table 1, and have been implemented for thermal monitoring in power grid systems and industrial plants operation [16]. OBS techniques include IRC, TSOP, and FBG approaches for junction temperature measurements. Infrared imaging using IRC is the early optical-based technique for capturing surface temperature distribution. In addition, IRC is still serving as a secondary measuring tool in most applications where other techniques are used for validation, thanks to its ability to quickly map the temperature distribution of a target surface from a distance [17]. The discovery of luminescence characteristics of semiconductors in forward bias in the 1990s was the primary drive behind the exploitation of TSOP for semiconductor switching devices. This method involves setting this device in an operation region where photons are emitted based on the magnitude of junction temperature and current [18]. Recently, the advent of optical fiber sensing revolutionized thermal monitoring techniques in aerospace and power transmission systems applications. They exhibit less weight and space, with a thickness of a few tenths of a micrometer, and as such can be easily embedded in power electronic circuits. State-of-the-art FBG is an optical fiber with an inscribed grating at a particular Bragg wavelength, which reflects light at this designated wavelength. A change in temperature over the grating region, typically associated with power switching semiconductor devices, alters the reflected Bragg wavelength utilized as a monitoring parameter to characterize the thermal behavior of the circuits [19,20].

Unlike the previous reviews that discussed in-depth electrical-based techniques [10,21], this review exclusively concentrates on OBS techniques and comprehensively discusses the three approaches viz. IRC, TSOP, and FBG. In particular, their underlined principles, recent advances, and comparison are presented. Moreover, this work also provides calibration and measurement guidelines for each of the OBS techniques, and finally, some possible ways to navigate through the open research opportunities that are identified to substantiate their practical implementation in the industry. This work is organized as follows: Section 2 discusses the structure and factors influencing junction temperature in power-switching semiconductor devices. Section 3 highlights different approaches to the OBS techniques based on IRC, TSOP, and FBG. Section 4 explicitly discusses the calibration and measurement setup for each of these approaches. Finally, Section 5 discusses some possible future developments and the implementation of FBG for commercial power electronic applications.

## 2. Power Semiconductor Devices

This section discusses the basic operational features, internal structure, and thermal behavior of power switching semiconductor devices. Typical semiconductor devices in power electronics include; thyristors, Silicon (Si)-controlled rectifiers (SCR), IGBT, and SiC MOSFET. While Si MOSFET operates at a high frequency and a low power range, Si IGBT is used in high power and low power to moderate frequency applications [22,23,24]. Recently, composite SiC MOSFET has been shown to exhibit low switching loss compared to Si MOSFET, combining the features and benefits of both Si IGBT and Si MOSFET, thus strengthening the potential of SiC MOSFET for high-frequency and high-power applications [25,26,27,28]. A comparison of all three popular semiconductor devices in terms of operating power and frequency of operation is shown in Figure 2. Moreover, the fact that Si IGBTs and SiC MOSFETs are the most frequently used power-switching semiconductor devices in several applications such as data centers, automotive systems, and power grids/systems, this review exclusively concentrates on both of these devices while illustrating their schematic diagram in Figure 3.

One of the fundamental lifecycle evaluation factors of power switching semiconductor devices is their junction temperature and its fluctuation, since this affects their lifetime, which may cause device failure [29]. Junction temperature refers to the mean surface temperature on the SiC MOSFET chip or the absolute maximum temperature of the emitter metallization on the Si IGBT chip. It is influenced by several factors. For instance, in a multilayer IGBT module that handles a wide range of input supplies, any random input voltage fluctuation causes the module to repeatedly hold up the thermal cycle’s shock for an extended period. Thus, junction temperature also fluctuates during this thermal cycle, giving rise to alternating thermal stress. Similarly, for SiC MOSFET, thermal stress influences the junction temperature variation due to high switching frequency [30]. In general, when there is a degradation in electron mobility, a further increase in power generation will also increase the junction temperature of these devices due to power dissipation. On the other hand, the aging of the solder layer can also contribute to an increase in thermal resistance, which in turn raises the junction temperature of the power switching semiconductor chips [31,32]. The internal structure of these semiconductor devices and the comparison are discussed in the subsequent paragraph.

Considering Figure 3a, the trench gate structure of Si IGBT that runs through the *n^+^*-emitter and *p*-base regions facilitates an increase in the channel density and eliminates the usual channel voltage drop inherent to junction MOSFETs. Moreover, the IGBT chip thickness is reduced by introducing an “*n*-fieldstop” layer that lowers the static and dynamic losses. Conversely, the conventional planar structure of SiC MOSFET, illustrated in Figure 3b, has the *n^+^* substrate region in contact with the drain electrode at the bottom of the device instead of the collector. In contrast to the structure of the IGBT, the emitter is replaced with the source electrode, while the gate electrode remains separated at the top by the interlayer insulator without a trench. Like Si IGBT, the channel of SiC MOSFET is located in the p region, between the *n^+^* source and the *n*-layer. Although both devices’ structure is similar to the MOS-gated structure, there is no parasitic body diode in Si IGBT, and thus it requires an antiparallel Si *p-i-n* freewheeling diode for practical applications [33,34].

From the electrothermal behavior viewpoint, both devices are conducting to the top or bottom surface of the die when there is a current flow. This causes variation in the temperature distribution within the device, and thermal modeling of these devices under the same current and voltage rating has shown that the junction temperature and the temperature swing of IGBT are higher than that of SiC MOSFET, since the on-state resistance of the IGBT is independent of junction temperature [35]. Meanwhile, in case of short circuit failure, the junction temperature rises faster in SiC MOSFET than in IGBT, which results in a lower short circuit holding time. This is because the heat generation rate in SiC is three times higher than the conduction rate, when compared to Si IGBTs. Hence, during a short circuit, junction temperature will be dominated by the heat generation rate [36], suggesting that the magnitude of the junction temperature for both devices depends on their operation state. The structure of both devices is similar to traditional MOSFET, and since both are Si-based semiconductors, they are suitable for the TSOP junction temperature sensing approach, as discussed in the subsequent Section 3. Moreover, since FBG could be bonded on SiC and IGBT devices while IRC can detect temperature distribution on their respective surfaces, both allow measurement of device junction temperature, thus making the OBS technique an attractive technology.

## 3. Junction Temperature Optical Sensing Techniques

The two physical-based techniques, thermistor and thermocouple, shown in Figure 1, have wide temperature measurement ranges and are readily available on the market. However, they suffer from the severe constraint of the mechanical process involved, which includes setting up and disassembling or making dents through the device, to enable probe contact with the chip. Conversely, the electrical-based methods exhibit fast response time and directly indicate junction temperature. Despite the fact that extensive calibration is required for each power circuit during its setup phase, the junction temperature estimation provided by TSEP and TSED is an average measurement [37]. Unlike optical-based techniques that employ light signals for temperature estimation, it is worth noting that both physical- and electrical-based techniques operate on the electrical signal, which is prone to loss due to the self-heating of the measuring devices. As such, both are invasive to the measurement or require additional external circuitry for compensation, increasing the power circuit’s complexity. The prominent features, advantages, and limitations of the three temperature sensing techniques are summarized in Table 1.

Despite several OBS techniques, such as Raman spectroscopy [38,39], liquid crystal thermography [40,41], and thermo-reflectance [42,43], being presented in the literature, only a few have been implemented for junction temperature sensing of power switching semiconductor devices. Although Raman spectroscopy and liquid crystal provide good resolution and are contactless to the targeted surface, they are impractical for large surface temperature measurements, since faster scanning is required [44]. Moreover, they are not commercially attractive due to the set-up complexity and cost of implementation. The three most popular OBS techniques commonly engaged for junction temperature measurement in power switching devices are the TSOP [45], IRC [46], and FBG [47], as highlighted in Figure 1. Unlike electrical and physical sensing techniques that are invasive to the system, OBS techniques are spatially separated from the sensing circuit and the device, since they operate on light signals. As such, they are immune to induced electrical noise and EMI from the surroundings. In the following, the underlying working principle, advancements in performance, and comparison of these three methods are discussed.

### 3.1. IRC Sensing Technique

Objects spontaneously emit radiation whose intensity and spectrum are temperature-dependent. Moreover, they also absorb, reflect, or stimulate emission to interact with the incident radiation [15]. The IRC sensing method exploits this naturally emitted infrared radiation from objects and hence does not require physical contact with the semiconductor chip surface [48]. It employs an infrared portion of the electromagnetic (EM) spectrum to sense the surface temperature through the emission of radiation. This technique can provide real-time temperature measurement, enabling the quick scanning and acquisition of stationary and fast-moving objects [49]. Generally, heat is emitted from the surface of the SiC/IGBT chip as infrared radiation and transformed into electrical signals via an infrared sensor. These signals are then mapped and displayed as a function of temperature in two-dimensional (2D) space for visualization purposes [37]. The main components of a typical IRC-based system are depicted in Figure 4. The IR detector located at the front end of the camera records the spectral emittance coming from the object (IGBT), which is then amplified and transformed from analog to digital data to generate a legible 2D thermal map. In some circumstances, postprocessing of the camera signal with signal processors is recommended for emissivity adjustment.

The 2D thermography mapping of the targeted surface temperature provided by IRC allows quick detection of hot spots during the measurement. However, the approach requires a clear line of sight between the camera and the target surface. Since power switching semiconductor devices are usually embodied with ceramics or plastic, a direct junction temperature measurement with IRC is challenging. Moreover, embedding IRC within the chip is not possible, owing to its substantial size and weight. Hence, in reported experimental works, the outer cover of the chip [51], or the die encapsulation [52,53], is usually removed to enable IRC to map the thermal distribution of semiconductor devices situated on a circuit board. In addition, a clear and fixed path is also established between the chip and IRC by fixing their position to obtain accurate measurements.

The infrared region of the EM spectrum spans up to 100 µm, but the range for temperature sensing in IRC is limited to 0.7–20 µm due to the reduced sensitivity of the IR camera’s photosensitive material above 20 µm [54]. As expressed in Equation (1), the emitted radiation striking the IRC is the function of the target material temperature, the atmosphere, and the radiant energy. Apart from the wavelength constraints mentioned earlier, the IRC approach also suffers from an exponential increase in noise with the rise in ambient temperature [55,56] and may also be affected by the variation in the measuring distance and angle [57,58,59].

The total radiation $W_{total}$ captured by the IRC, considering the emission from the atmosphere and surroundings, in addition to the emission from the object, as shown in Figure 5, can be expressed as: (1)$$W_{total}=E_{obj}+E_{atm\;\;\;}+E_{refl\;}$$
where $E_{obj},$ $E_{atm\;},$ and $E_{refl}$ are the emissions from the target surface, and reflections from the atmosphere and the surroundings, respectively. Moreover, these emissions can further be expressed as: (2)$$E_{obj}=\varepsilon_{obj}\;\;\tau_{atm\;\;\;}\sigma_{atm\;\;\;}{\left(T_{obj}\right)}^{4}$$
(3)$$E_{atm}=\varepsilon_{obj}\;\;\tau_{atm\;\;\;}\sigma_{atm\;\;\;}{\left(T_{atm}\right)}^{4}$$
(4)$$E_{refl}=(1-\varepsilon_{obj})\;\tau_{atm\;\;\;}\sigma_{atm\;\;\;}{\left(T_{refl}\right)}^{4}\;$$
where $\varepsilon_{obj}$ is the emissivity of the target surface (i.e., the object), $T_{obj}$, $T_{atm}$, and $T_{refl}$, are temperatures of the object, atmosphere, and reflection, respectively, and $\tau_{atm}$ is the transmittance of the atmosphere. $\sigma_{atm}$ is Stefan-Boltzmann’s constant, which is given as 5.670 × 10^−8^ W/m^2^/k^4^. Emissivity, which refers to the ability of an object to emit thermal energy [60,61,62], dramatically affects the measurement accuracy of the IRC sensing technique [43]. For the IRC-based system to obtain accurate thermal measurement, emissivity of the targeted object must be uniform. For instance, commercial IGBT chips are usually coated with silver solder layers by manufacturers, which significantly decrease its surface emissivity. As such, the surface temperature of the IGBT chip could not be correctly measured by the IRC as reported in [63]. Nevertheless, we can characterize the surface emissivity considering the blackbody and infrared radiation discussed in [64] as:(5)$$\varepsilon_{obj}=\frac{\left(R_{T1}-\;R_{T2}\right)}{\left(R_{b1}-\;R_{b2}\right)}$$
where *R*_*T*1_ and *R*_*T*2_ are the infrared emission levels at known temperatures, *T*1 and *T*2, and *R*_*b*1_ and *R*_*b*2_ are the equivalent black body emission levels. The advancements in this IRC optical sensing are summarized in Table 2.

To increase the surface emissivity, Baker et al. [64] employed filtered paint and micro-spraying equipment to spray the target surface to emphasize further the requirements for improved surface emissivity for reliable junction temperature measurement of an IGBT chip. They ensured that the particle size and paint thickness (<100 µm) were uniform throughout the surface, which they showed was necessary to achieve homogenous surface emissivity. Another approach for fixing the emissivity is to use signal processors to directly postprocess the output signal of an infrared camera, as shown in Figure 4. Although the approach enhances the complexity of IRC, it has the benefit that it can map the temperature of any semiconductor chip, independent of its shape or composition [50].

As indicated in Table 2, it was observed that the experimental results obtained in [65,66] have a temperature error of ±0.12 K at a uniform emissivity compared to cases where emissivity falls below 1, such as ref. [67]. In this case, a temperature difference of ±1.25 K at an emissivity factor of 0.735 has been reported within a temperature range of 100 to 160 °C. This suggests that the effect of emissivity cannot be overlooked. Furthermore, frame rate and spatial resolution are also essential factors in evaluating the performance of IRC. The frame rate is the speed at which the camera updates the temperature readings to be displayed on the screen. An IRC with a high frame rate is desirable to capture rapid temperature changes. Due to the limited refresh rate of 9 Hz exhibited by the IRC used by Cheng et al. [68], the junction temperature variation could not be captured when the applied pulse width modulation (PWM) was raised above 10 Hz at the gate of IGBT.

**Table 2 micromachines-14-01636-t002:** Advancements in IRC sensing technique for junction temperature measurement of switching semiconductor devices.

Year	Device	F_S_ (Hz)	Pixel	C_TH_ (µm)	ε	ΔT (°C)	Err	Ref.
2021	SiC M.	200	-	-	0.73	100–160	±1.25 K	[67]
2021	IGBT	9	-	-	-	39–100	-	[68]
2020	IGBT	200	-	-	~1	70–140	±3 °C	[69]
2017	IGBT	100	-	10–16	-	80–190	±2 °C	[70]
2016	IGBT	160	-	-	~1	30–80	±5 °C	[71]
2016	IGBT	50	-	-	-	35–60	3.4%	[72]
2016	IGBT	106	48 × 4	-	-	25–50	-	[73]
2013	IGBT	-	1368	100	1	40–180	±0.12 K	[65]
2012	IGBT	100	-	-	0.94	70–140	<3%	[74]
2010	IGBT	50	320 × 256	-	~1	40–70	±2 °C	[75]
2007	IGBT	15	-	220	~1	35–60	±1 K	[66]

F_S_: frame rate, C_TH_: coating thickness, ε: emissivity, ΔT: temperature range, K: Kelvin, SiC M: Silicon Carbide MOSFET, Err: measurement error.

An IRC of sampling rate above 100 Hz, employed in [69,71], reported capturing transient temperature across the junction with an accuracy of ±3 °C in the temperature range of 70 to 140 °C. The existing costly IRCs available in the market exhibit frame rates up to 200 Hz with better scanning speeds. Nevertheless, the accuracy and transient temperature measurement capability could be improved. In this case, the IRC images are sampled once a steady thermal state is reached. Some of the established issues with IRC include surface reflectance and uncertainty in local temperature. The probable reason for the significant change in the emissivity was presented as the transparency of the molded lens and reflection from other components of the targeted object [37]. This issue could be mitigated by removing objects around the setup that may likely cause reflection; otherwise, the surface could be painted black. 

The uncertainties of local temperature due to variation in surface emissivity are due to changes in radiative recombination properties of semiconductor layers, bonding elements, and coating having different transparency or reflectance properties to infrared radiation, making it difficult to determine the surface emissivity accurately. These issues have been mitigated in the literature with high emissivity coatings [50,76]. Another alternative to obtain an accurate measurement of emitted radiation from a target is to place micro carbon particles near the targeted surface, eliminating the need for coating [77]. It should be noted that IRC may provide nonuniform temperature estimation if the thickness of the coating varies across the surface. 

Other factors to consider for the accuracy of the IRC sensing technique, including the spatial resolution and image pixels, which defines the ability to provide details of temperature distribution on the captured surface, which relies on the pixels of the camera’s detector and its field of view (FOV) specification with respect to the area the camera sees at any given moment [53]. The spatial resolution of the IR image could be defined with an adapted lens to permit pixels of various sizes and FOV, which was utilized in [75] to improve the accuracy of the IR camera to about ±2 °C. Also, the influence of system noise and the error during detection could be compensated with auto-calibration and image-processing algorithms embedded in commercially available IR cameras.

The summary of benefits and challenges provided by this IRC OBS technique is shown in Table 3. As highlighted before, IRC is attractive to evaluate power electronics circuits’ thermal validation, reliability, and temperature performance at a designated distance only when there is a clear path to the targeted object. 

### 3.2. TSOP Sensing Technique

The TSOP sensing technique uses the electroluminescence (EL) phenomenon to measure the junction temperature of Si power switching devices. As shown in Figure 6a, the radiation emitted (blue-like visible light around SiC chip) as a result of external stimulation, such as an electric field or photon excitation, is known as luminescence [78,79], and EL, in particular, is the emission that occurs as a result of excitation through the recombination of electrons and holes across a *p-n* junction, or in other words, via bias voltage [80]. The peak energy of luminescence depends on the temperature and the spatial resolution of the EL [15], while the spatial resolution is dependent on the area of the *p-n* junction that is producing the light signal [79,81,82,83]. 

The EL process includes simultaneous current and temperature effects. A controlled source is used to generate current in order to control the conductivity of the body diode of the SiC MOSFET. The light emitted from the SiC is then exported to an optical grating spectrometer for spectral analysis via a quartz optical cable fixed to the MOSFET chip. The heat controller positioned beneath the SiC accounts for the junction temperature difference at various forward current settings to decouple the effect of the temperature and forward current on the emitted photons. The spectral features exhibited in this process can be adapted to estimate the junction temperature of other semiconductor devices such as LED and Si MOSFET [84,85]. Figure 6b shows a generic schematic of the TSOP technique for junction temperature measurement. The optical path is typically a low-loss optical fiber sensor for transmitting EL to the spectrometer for spectral analysis. The fiber sensor tip is often fixed onto the Si chip die for EL extraction via optical power coupling to the fiber.

Generally, every forward-biased *p-n* junction semiconductor device can emit light, and this luminescence will be strong for direct bandgap semiconductors. Unfortunately, power switching devices, such as SiC MOSFETs and Si IGBTs, are indirect bandgap *p-n* junctions; thus, the radiative recombination process is very weak [80,86]. However, they can act as parasitic light-emitting diodes while they are forward-biased, thus emitting weak EL, which makes it possible to detect and hence it can be utilized as a viable sensing parameter for Si/SiC power switching devices. Moreover, for EL to be strong, SiC MOSFET and Si IGBTs are operated in the third quadrant window where the forward-biased condition can be achieved [87]. The measurement of photoemission bandgap carried out in [88] has shown that 4H SiC, the most-used SiC polytype in power electronics due to its thermal and mechanical properties, has a junction emission in the ultraviolet (UV) spectrum with a −3.62 eV energy band. 

For SiC MOSFET, the body diode only acts as a parasitic light-emitting diode during passive third quadrant (forward bias) operation [89], in other words, when the MOSFET is in the OFF state (i.e., the gate-source voltage *V_gs_* is nearly zero, and the current flow is in the opposite direction to the device). In the case of Si IGBT, the *p-n* junction near the collector is forward-biased, and a collector current *I_c_* flows into the device when it is in ON state. As a result, its *p-n* junction acts as a parasitic light-emitting diode in the first quadrant operation. To establish third quadrant operation, antiparallel freewheeling diodes are usually included with IGBTs chips. Since the light emission from power semiconductors is obtained during the passive third quadrant operation, the measurement period is limited to the dead time in power electronic applications [90,91].

The spectral sensitivities of SiC MOSFET’s EL are influenced by the gate bias voltage and bias temperature instability (BTI). Both alter the effective electrical field across the oxide and cause changes in the current that flows through the body diode, which, in turn, impacts EL extraction [92,93,94]. To mitigate this, Lukas et al. [95] proposed a post-processing method of minimizing SiC’s sensitivity to gate bias voltage using the estimated intensity ratio of both spectral peaks (UV and blue-green peaks). Experimental work in [96,97,98] has established that two notable spectrum peaks are significant in a typical SiC EL spectrum. As shown in Figure 7a, the UV peak centered at ~390 nm due to band-to-band recombination, and the blue-green peak centered at ~500 nm owing to the recombination of deep Boron states with the conduction band and acceptor states caused by doping elements and lattice impurities [99,100,101]. Since the remaining energy in the recombination process is released in the form of a photon, the relationship between the emitted light peak wavelength *λ* (nm) and photon energy *E* (eV) is given as: (6)$${\;\lambda \;(nm)}_{}=1240/E\;(eV)$$

Since the energy bandgap of semiconductors is temperature-dependent, the spectral power distribution properties of Si and SiC, and other features such as the peak wavelength and spectral bandwidth, can be used to characterize junction temperature variations [102,103]. The experimental work in [104] investigated the EL of SiC MOSFET, and developed an electrothermal-optic model that showed the relationship between the EL intensity, junction temperature, and forward current, which is given in the form of total light intensity *I_EL_* as:(7)$$I_{EL}=a_{o}i\left[1+\;k_{1\;}\Delta T\right]+b_{o}i^{2}\;\left[1+\;k_{2\;}\Delta T\right]\;\;\;\;\;$$
where *a_o_* and *b_o_* are the coefficients due to the effect of junction temperature, *i* is the forward current, $k_{1}$ and $k_{2\;}$ are the constant coefficients, and ΔT is the change in the junction temperature of the device. Thus, at a given forward current condition, light intensity varies with respect to junction temperature. Moreover, the device output voltage $V_{0}$ is not only a function of $V_{T0}\;$(initial junction voltage) but also the temperature change, and is expressed as:(8)$$V_{0}=V_{T0}\left[1+\;k_{1\;}(T_{0\;}-T_{j}\right]\;\;\;$$
where $T_{0\;}$ and $T_{j}$ are the given temperature at $V_{T0}$ and the junction temperature, respectively. Thus, the expressions for $I_{EL}$ and ${\;V}_{0}$ in Equations (7) and (8), show that junction temperature varies linearly with an increase in light intensity extracted from the device, corresponding to the spectral waveform as shown in Figure 7a [105]. 

A typical implementation of TSOP employs a low loss nonlinear fiber optic sensor for transmission and, in most cases, has the sensor tip fixed on the Si chip die for EL extraction via optical power coupling to the fiber, as illustrated in Figure 7b. The visible light emission around the decaped SiC chip during the conduction of the body diode indicates the presence of inherent electroluminescence in the SiC body diode. Also, the variation in the output voltage of the photosensitive sampling circuit corresponds linearly to the rise in the chip’s junction temperature, as expressed in Equation (8). However, the observed EL emission is weak, since the radiative recombination in SiC is low due to the dominant non-radiative combination.

**Figure 7 micromachines-14-01636-f007:**
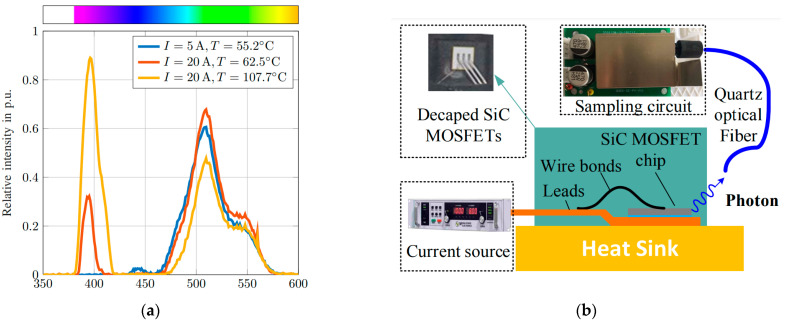
(**a**) Electroluminescence spectrum of a SiC diode for different device currents and junction temperatures Reprinted with permission from Ref. [96], 2023, IEEE. (**b**) Experimental setup for TSOP technique extraction from SiC MOSFET Reprinted with permission from Ref. [105], 2023, IEEE.

The approach in [106] utilized the TSOP technique for measuring the junction temperature of two paralleled SiC MOSFETs. Here, the extraction of light emission from individual chips was carried out independently. The two junction temperatures, *T_j1_* and *T_j2,_* were obtained from the integrated intensities of the sub-peak areas. This ensures accurate estimation via optical fibers connected to the optical spectrometer, since the module’s current is not evenly distributed. Thus, the light transferred by each fiber to the sensing circuitry depends on the individual temperature and current. Table 4 summarizes the recent advancements in the TSOP technique from the literature.

Another approach is using EL spectral features to extract the junction temperature and current at once, as shown in Figure 8. This was achieved with multiple optical fibers and TSOP sensors exhibiting different wavelength sensitivities, and was further processed by an artificial intelligence system [83,96]. The advantage of this method is that the junction temperature and device current could be estimated individually from the paralleled devices. However, the time instant of extraction could not be selected simultaneously to light emission, since all SiC MOSFETs of the power switching circuit do not emit light simultaneously. EL extraction from SiC MOSFET has also been implemented in high voltage applications, such as traction inverters, using a Si photomultiplier [87]. A repetitive 50 ms pulse current was applied to obtain emission, which was detected by a fiber-coupled *p-i-n* photodiode. The spectrum exhibits a significant characteristic peak around 500 nm, while the intensity–current characteristics related to the temperature coefficient, obtained at −0.003 V/K, were utilized to estimate the variation in the junction temperature. Similarly, in [105], this method was used but at a temperature coefficient of −0.0046 V/K and a sensitivity of 3.2 mV/K between the temperature range of 30 to 150 °C was reported.

Due to the weak EL signal in the TSOP sensing technique, an integrated operational amplifier was engaged to improve the signal-to-noise ratio (SNR); nevertheless, a slight deviation in measurement accuracy was observed as the junction temperature rose due to the self-heating effect. Moreover, the galvanic isolated sensing method for SiC MOSFET was introduced in [108] based on the variation in light intensity. The extracted EL spectrum exhibited two characteristic peaks around ~380 nm and ~480 nm, while a similar approach implemented in [104] has ~383 nm and ~485 nm for the two peaks, as indicated in Table 4. The two peaks exhibited different temperature coefficients. The major peak had a negative temperature coefficient, while the minor one showed a positive peak. The emission was coupled to the spectrometer via a quartz fiber fixed on the side of the chip for measurement purposes. The former used two independent bandpass filters to extract the peak emissions, and the signal ratio was used to compensate for the EMI due to fast switching transients and optical transmission degradation within a temperature range of 30 to 150 °C. On the other hand, the latter work displayed a limited temperature range between 90 and 135 °C, and the sensitivity obtained was ~1.53 mV/°C at a mean error of ±5 °C; the optical measurement, in this case, still required a high bandwidth since the dead time was short.

The approach for extracting the forward current and junction temperature from the two peaks simultaneously was implemented by integrating inherent electrical isolation with the EL technique [107]. After establishing a correlation between the current and temperature through an analytical model, a negative gate voltage of −5 V was applied to the gate of SiC MOSFET to operate in the third quadrant. The spectral shifts of the two peaks at 510 nm and 390 nm were used to determine junction temperature at a mean error of ±3 °C within a temperature range of 50 to 130 °C. A similar approach was also implemented in [96] with similar major and minor spectral peaks, but an improved sensing performance was reported with an error of ±1.2 °C in a temperature range of 10° to 90 °C and using multiple sensors for photodetection. However, this approach employed a high-resolution spectrometer for measurement, since the peak wavelength shifts noted were only a few nanometers. In addition, optical measurement with a high gain was required since the emission from the body diode of SiC is not very efficient due to its indirect bandgap.

Light-sensing circuits such as a photodetector typically detect the spectral characteristics of light emission through EL. In the case of multiple parallel devices, multiple detectors of different wavelength bands may be implemented for spectral sensitivity; otherwise, the guided light has to be filtered for proper detection [83]. Also, a major limitation of EL application in SiC MOSFET is that the chip is packaged in plastic and covered by metal on the top, which makes it challenging to obtain a direct measurement. When a direct current is applied to the body of diode, the light emitted around the chip cannot be fully extracted. Thus, the ratio of weighted spectral information obtained by the photodiode does not reflect the actual junction temperature. In addition, the cross-sensitivity of junction temperature and current is another challenge in SiC MOSFET TSOP sensing.

Although TSOP is not as sensitive as a thermal camera, they are still usable for online junction temperature measurement and have been implemented for SiC MOSFET in the literature. Other challenges related to the setup are discussed in Section 4 of this article, while the summary of this technique’s key advantages and challenges is highlighted in Table 5.

### 3.3. FBG Sensing Technique

The thermography method gives average temperature distribution over an area and thus is incapable of detecting maximum temperature for a targeted point [111]. FBG is a very recent optical technique explored in the literature for measuring the junction temperature of power switching semiconductor devices. The working principle of FBG is based on the wavelength shift that occurs because of the variation in thermal profiles over the grating portion of the fiber sensor, and the wavelength properties can be characterized to represent the variation in junction temperature in SiC MOSFET and Si IGBTs. In a typical FBG, the manufacturer engraves the Bragg gratings into a single-mode fiber. When the light signal from a broadband source is fed to the fiber, the light signal known as the Bragg wavelength is reflected, depending on the refractive index and the grating period [112] and is given as:(9)$${\;\lambda }_{B}=2\;\;n_{eff}\;\Lambda$$
where Λ is the grating pitch and $n_{eff}$ is the effective refractive index of the single mode fiber. Whenever the external temperature around the grating portion of the fiber is varied, the thermo-optic effect alters the core refractive index [111], thus affecting the $n_{eff}$, which shifts the central wavelength of the reflected signal accordingly. The relationship between the temperature variation and the wavelength shift of the FBG sensor is given by: (10)$$\frac{\Delta \lambda }{\lambda }=\left[(+\;a_{f\;}+\;\xi )\Delta T\right]+(1-\;P_{e\;})\Delta \varepsilon$$
where Δλ is the wavelength shift, *λ* is the initial center wavelength of the FBG, $a_{f\;}$ is the thermal expansion coefficient of the fiber, *ξ* is the thermo-optic coefficient, and $P_{e\;}$ is the strain-optic coefficient. Hence, Equation (10) considers wavelength shift due to the temperature and strain; however, in the case of junction temperature measurement, only the effect of temperature is required. The external strain on the FBG can be eliminated by using a tube or rigid housing, which additionally protects the sensor from mechanical damage [47]. In this case, Equation (10), can then be rewritten as:(11)$$\frac{{\Delta \lambda }_{b\;}}{\lambda_{b\;}}=\;\;(a_{f\;}+\;\xi )\Delta T$$

A typical measurement flow chart of the FBG sensing technique is shown in Figure 9, The setup involves bonding the FBG sensor to the semiconductor chips, usually inside modules, with thermal oil or glue to improve the thermal contact between the chip’s surface and the fiber sensor [29,113]. The fiber is then illuminated with a broadband light source (BBS) via an optical circulator, and the sampling rate is pre-selected depending on the interrogator type. The reflecting light signal wavelength shift can then be routed to the interrogator and examined for various pulse widths applied to the Si IGBT gate. Based on the predetermined sensitivity, the junction temperature due to conduction losses for long-duration pulses can be computed. However, the transient losses during switching and the peak power generated during ON-OFF switching losses may or may not be detected depending on the interrogator acquisition rate. Table 6 depicts the recent advancements in FBG sensors in the junction temperature sensing of power switching semiconductor devices. FBG sensors can be enclosed in a tube to improve the shear strength of the sensor; it should be calibrated to minimize measurement errors that may be introduced due to environmental factors. A temperature-wavelength relationship can then be obtained for the FBG sensor with the least square method (better fit with a linear equation of about 99.9% precision), and an accuracy of up to ± 1 °C was reported at a temperature below 40 °C in [111].

The effect of airgap and the interrogator sampling rate was studied in experimental work by Liu et al. [112]. A pulse of different widths and duty cycles was applied at the gate of the MOSFET while the oscilloscope measured the input direct current and voltage. The interrogator monitored the wavelength shift to capture the thermal pulses and the thermal sampling frequency was limited to 1 kHz; thus, the optical interrogator captured changes in wavelength every 1 ms. It was observed that the junction temperature increased and accumulated as the duty cycle increased from 2 up to 50% for the same pulse width of 300 s. However, when the pulses ended, a slight temperature rise was initially observed, after which the temperature dropped gradually due to the error introduced by the air gap between the grating portion of the sensor and the die, and the limited sampling rate of the employed interrogator.

Chen et al. [68] proposed direct on-chip thermal measurement for Si IGBT to demonstrate the effect of the sensor bond interface with the chip surface and evaluated the performance of real-time junction temperature measurements. For the direct detection of the die temperature, the ceramic package of the IGBT module was removed and the FBG sensor was placed directly on the chip, as shown in Figure 10a. Measurements from two FBG sensors with different interfaces (air, and solid bond interfaces with IGBT chips) were then examined. The solid-interface setup has the FBG sensor bonded to the chip interface with a thermal paste, which had a thermal conductivity of 5.2 W/(mK), exhibiting an accuracy of 2 pm (~0.2 °C) compared to that of the air interface (without thermal paste), which exhibited an accuracy of 3 pm (~0.3 °C). The air gaps between the fiber sensor and the chip influenced the sensor response. Moreover, the air-interface response exhibited a much slower rise rate than the solid-interface case when both were subjected to fast temperature changes. However, in [115], a groove was cut on the baseplate along its axial centerline, beneath the IGBT chips, instead of on-chip placement, as shown in Figure 10b. This method allows the embedding of FBG sensors in the device module without interfering with the IGBT operation, although thermal network characterization of the module is required to obtain equivalent junction temperature. 

The experiment by Ren et al. [47] investigated the effect of packaging schemes on temperature sensitivity and transient performance. Three FBG sensors, a bare FBG, a metallic plate housing, and a tube housing were used for temperature sensing under various conditions. For FBG with metallic plate housing, the sensor was not segregated from the plate and the paste. As such, it has the drawback of allowing the strain on the plate to spread further to the FBG sensor, which causes errors in the measurement. Thus, the device was calibrated after inserting the sensor in the housing to take care of the effect of strain. From the experimental results, bare FBG with no packaging had a sensitivity of 10.2 pm/°C, while the tube and plate housings were recorded 10.4 pm/°C and 14.7 pm/°C, respectively. It was observed that there is a significant increase in the sensitivity of plate housing to about 1.45 times bare FBG. At 10 Hz, the plate housing type effectively detects the temperature ripples at different frequencies, having peak-peak ripples at 2.58 °C. Nevertheless, the tube housing could not capture the temperature ripples due to its slow response. Thus, it was established that the plate housing type is the preferred packaging method for chip transient temperature measurement as it captures intercycle temperature ripples at several modulation frequencies. 

The influence and performance of FBG for different grating head lengths were tested in reference [114]. It was suggested that uneven heat distribution of the temperature captured by the FBG sensor might result from unsuitable head dimensions. To investigate the effect of head dimensions, three FBG sensors with different head lengths of 1 mm, 3 mm, and 5 mm were assessed in the experiments. To ensure consistency of comparison and assessment of length effects, the midpoints of all three FBG head lengths considered were kept at identical locations during the test. Also, the measurement for each of the fibers was taken independently. Compared with the simulation, the obtained temperature measurement for 1 mm FBG was approximately the same within a range of 45 °C. In contrast, the longer 3 mm FBG sensor variations were within 1.6 to 1.9 °C of the actual measurements, while the 5 mm FBG displayed around 5.2 to 6.0 °C lower than the results obtained for a 1 mm FBG head sensor. This is equivalent to a 16% relative deviation from the desired temperature values. As such, it is unacceptable in a situation where hotspot precision is required. As is evident from the results of the above experiments, FBG with short head lengths are preferred as they provide more accurate detection of localized hotspots. However, FBG with a short head length has the drawback of ensuring precise location and sensor placement. In contrast to short FBG sensors, longer FBGs can deliver accurate temperature readings in areas with less extreme thermal gradients and favor placement with simple installation.

The key benefits offered by FBG sensors, along with underlying challenges, are summarized in Table 7. So far, sampling rate, grating head length, and thermal conductivity are major factors influencing FBG accuracy, which depend on the manufacturer’s specifications. Error due to sensor housing can be controlled with proper calibration. Smaller housing is necessary for compatibility, since FBG sensors will be incorporated into the power module. In addition, compact housing will allow quick response, which is also necessary for transient temperature measurement. The advantage offered by FBG over other optical-based techniques, if embedded adequately in MOSFET and IGBT modules, provides fast and accurate measurement at a low cost compared to the thermography technique counterpart.

### 3.4. Summary

The three popular OBS techniques, IRC, TSOP, and FBG, discussed so far, have been extensively tested in the literature for SiC MOSFET and Si IGBT, and are reviewed in Table 2, Table 4 and Table 6, respectively. They are noninvasive and offer accurate measurements if professionally installed and calibrated. Furthermore, it is worth noting that the FBG-based techniques require low attenuation fibers with negligible bending radius and insertion losses (i.e., connector losses) and hence should be considered during installation. A comparison of various characteristics of the three OBS techniques is shown in Table 8, suggesting that no single approach excels over others. Hence, the selection of the technique invariably depends on the application and the surrounding constraints.

## 4. Method of Calibration

This section discusses the materials and apparatus needed for calibration and the experimental setup of the three key OBS techniques for junction temperature measurement in power electronics applications.

### 4.1. IRC Sensing Technique

The calibration setup for thermal imaging consists of a lens with a predetermined working distance to focus the thermal radiation on the camera’s detector [37]. In the literature, the reported distance is between 15 and 25 cm. An adjustable emissivity setting is required to calibrate the detection based on the orientation of the targeted object and ambient conditions for each setup. 

As illustrated earlier in Figure 5, the infrared thermal imager should be positioned at a distance and to the front of the IGBT, as a direct line of sight is required; the focal length is firstly set, after which the camera is fixed with a gripper once the infrared thermal imager is correctly displayed on the screen [111]. For commercial IGBT, the dielectric gel on the chip surface may be removed and painted for uniform emissivity across the chip surface for accurate measurements. Also, the paint may be filtered to attain a uniform particle size whose thickness ranges from 5 to 16 µm, as suggested in [70], to improve the accuracy.

### 4.2. TSOP Sensing Technique

Proper calibration is essential for implementing TSOP methods, because the estimation of junction temperature by EL is based on intrinsic properties, which are sensitive to differences in fabrication and variations in electrical parameters that affect the operation of the MOSFETs in the third quadrant window [108]. For SiC modules mounted on a ceramic substrate and covered by a transparent silicone gel, the detector could be immersed in the silicone gel to enhance the optical coupling [105]. Otherwise, a photodetector may be mounted above the side wall of the device to measure the light emitted from the surface of the substrate. Wang et al. [19] placed the detector at about 2 mm over a SiC chip covered with transparent silicone gel. Considering that the light wavelength travels through the gel, a dark box shielded the ambient light from the setup desk to avoid disturbance from external noise.

Typically, SiC MOSFET is biased with a negative voltage within the range of −5 to −15 V to set the device in reverse conduction mode to achieve the third quadrant operation for emission. As illustrated earlier in Figure 7b, the SiC module is placed on a heat controller, which could be a digital hot plate designed to raise the diode’s temperature to decouple the influence of forward current and the junction temperature on the emitted light. The required pulse current for the EL process could be generated from a DC source in a typical range of 2 to 50 A. Depending on the desired accuracy and budget, a Si *p-i-n* photodiode can be used to detect photons with or without external bias. The photodetector generates a current proportional to the emission, depending on the junction temperature of the SiC MOSFET. In addition, the spectrum emitted under different conditions can also be analyzed on a spectrometer for characterization purposes [106]. 

### 4.3. FBG Sensing Technique

For the FBG method, the temperature wavelength fitting curve is necessary to determine the sensor’s sensitivity. Light can be fed into the fiber from a BBS to calibrate the sensor, while the reflected light from the sensor can be routed to the spectrum analyzer or an interrogator via an optical coupler, as shown in Figure 11. Next, a portion of the grating area is heated in an enclosed environment whose temperature can be precisely controlled to a predetermined value. Alternatively, a heating plate can be used together with a thermocouple to validate the temperature at the desired time. Next, to ensure the mechanical stress-free sensor, the FBG head could be inserted into a ceramic capillary attached to a stainless-steel plate using Kapton tape [114]. 

The initial temperature and central wavelength shift are first recorded, then the temperature is raised with a fixed step size, and at each value, the corresponding wavelength shift is recorded until the maximum value is reached. This procedure can be repeated several times to take an average value of the wavelength shift for the number of cycles and calculated at each temperature level. The data obtained can then be used to compute the temperature wavelength fitting curve using a linear regression (*y* = m*x* + c), where c is the initial Bragg wavelength at ambient temperature, *x* and *y* are the wavelength and temperature at each point, respectively, and m is the slope of the curve, which represents the sensitivity of the FBG sensor. The typical measurement from the literature has shown the slope of the linear fitting in the Kelvin scale to be 10.99 ± 0.073 pm/K, with a mean error of ±0.5 K [111]. 

In [75,115], the central Bragg wavelengths of the FBG sensors used are 1537 and 1539.9 nm, respectively, exhibiting a sensitivity of ±0.2 nm/°C. After the calibration, the value of the junction temperature estimation can be obtained by measuring the wavelength shifts. For practical measurements, the fiber can be bonded to the chip with a thermal paste of high viscosity and low shrinkage [111]. This is necessary to ensure improved heat exchange between the sensing portion of the fiber and the IGBT chip surface. The thermal conductivity of the thermal paste and the temperature must be factored in during selection to reduce the effect of aging and thermal breakdown [50].

## 5. Future Recommendations

### 5.1. Distributed Temperature Sensing

The OBS techniques explored so far have been designed for a point or single-unit junction temperature measurement. A typical commercial power circuit contains two to tens of IGBTs or SiC MOSFETs to capture the semiconductor devices, where distributed temperature sensing (DTS) could be a viable solution for such applications. Among the three optical techniques discussed in Section 3, the FBG technique is the most suitable approach for DTS of power switching circuits due to its size, maintainability, and overall cost compared to IRC and TSOP. As depicted in Figure 12, an array of IGBTs or SiC MOSFETs can be monitored concurrently with several FBG sensors on a single optical fiber cable. The fiber is carefully laid so that the sensing portions of the fiber grating are situated on the device’s chip. Each FBG sensor is uniquely identified based on its center wavelength so that once the light is allowed to pass through the fiber, each FBG sensor reflects at a designated Bragg wavelength based on the junction temperature of the device. Moreover, several sections of such fiber could be combined at the fiber flange. The fiber flange will provide an interface for connecting multiple fibers to form a distributed system for commercial applications; this interface could be made passive such that the light is transmitted, and the signal processing is shifted to the central monitor through an optical link such as free space communication (FSO). Otherwise, a photodetection circuit could be embedded to handle the data processing within its locality. 

### 5.2. Reduction in Response Delay

A slight delay in response is typical in IRC and FBG sensing techniques and is more prominent when the power module temperature falls. This is due to the equipment capacity limitation and the external system’s effect on the measurements. For IRC, reflections from the surroundings and target distance could be the causes. To mitigate this issue, an IRC with an adjustable sampling rate and pixel resolution could be a viable solution. For the FBG, this issue may occur due to the minute air gap between the sensor portion and the targeted surface (sensor-chip gap) and the limited acquisition rate of the interrogator. An interrogator or spectrum analyzer with an auto-adjustable acquisition rate will facilitate a quick response. In addition, the sensor-chip interface could also be matched as close as possible to the specially treated thermal paste, which can be uniformly applied across the surface to avoid future heat loss and chirping failure. 

### 5.3. Automated Calibration and Intelligent Operational Prognosis (ACIOP)

Among the temperature sensor technologies, the OBS technique and FBG approach have the notable benefit of exhibiting linear sensitivity. However, they require calibration, because the surrounding ambient temperature and sensitivity of the FBG sensors slightly differ from each other. Although auto-calibration for the IRC technique is now available on the market, this feature has not been exploited or implemented for FBG and TSOP approaches. Machine learning could be incorporated to fill this gap, using the current ambient temperature and pre-trained data to compute FBG sensitivity to automate the calibration process. This could be a promising way to eliminate manual calibration. ACIOP, in this context, could use a deep learning algorithm to calibrate and predict each unit’s junction temperature based on designated features such as the magnitude of the load across the power module, frequency of operation, usage, and components’ aging. The relationship established by the deep-learning model from the highlighted features could further be used to evaluate component reliability, life cycle, and usage under various operation states. To integrate these features into FBG techniques in the future, adequate data acquisition and data-driven models would be necessary for the model to provide an acceptable prediction under various conditions. 

## 6. Conclusions

A thorough comparative review of the state-of-the-art OBS techniques for junction temperature sensing in power switching semiconductors has been performed. It was established that IRC rendered a better 2D temperature mapping but could not be embedded in power electronic circuits. The TSOP technique, on the other hand, is simple to implement but only practically applicable for SiC MOSFET. The FBG technique exhibits high spatial resolution and compact size, which makes it attractive to be embedded in power electronic circuits. However, logical positioning and a suitable packaging method with a uniform shear strength are required to obtain accurate temperature measurements. With the rapid growth and deployment of optical fiber sensors for various applications, multiparameter and distributed temperature sensing should be considered to gain widespread use for commercial applications. So far, we have showcased the implementation of distributed temperature sensing and intercommunication for data acquisition to enable sensor integration to a central monitoring platform through a communication link. These areas needed more attention to facilitate the development of other necessary features, such as automatic calibration, to make FBG adaptable for other power electronic applications. Lastly, we highly suggest that researchers explore this domain in order to achieve industrial breakthroughs.

## Figures and Tables

**Figure 1 micromachines-14-01636-f001:**
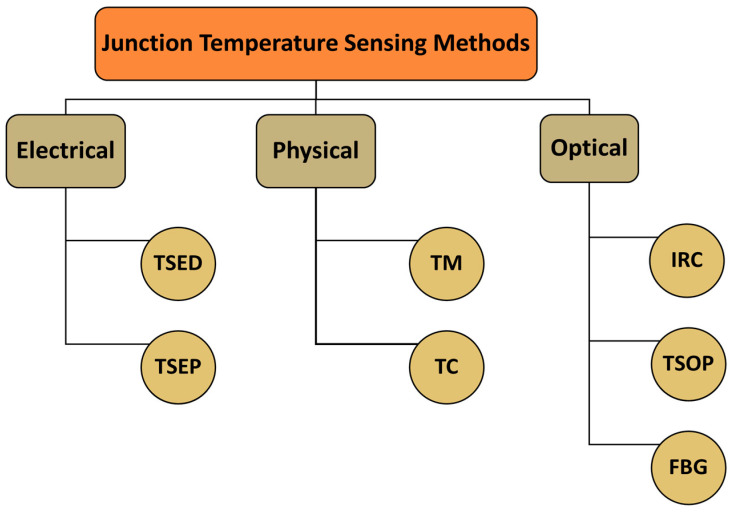
Classification of junction temperature sensing techniques for power switching semiconductor devices.

**Figure 2 micromachines-14-01636-f002:**
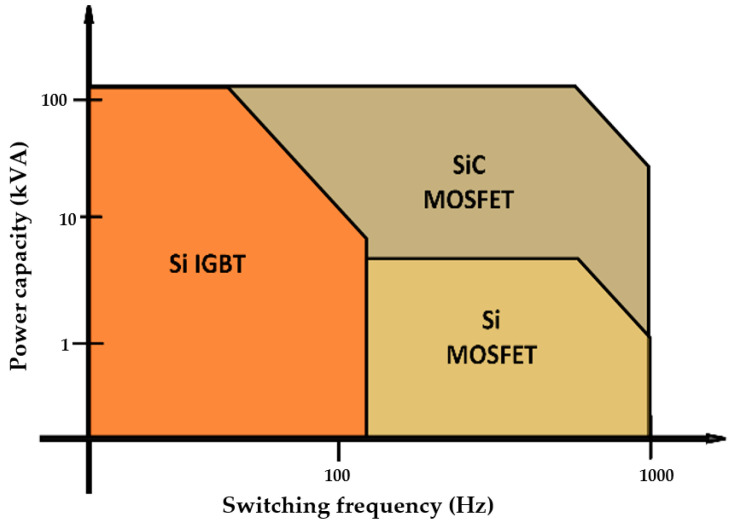
Comparison of the operating power and frequency for Si and SiC transistors (IGBT and MOSFET).

**Figure 3 micromachines-14-01636-f003:**
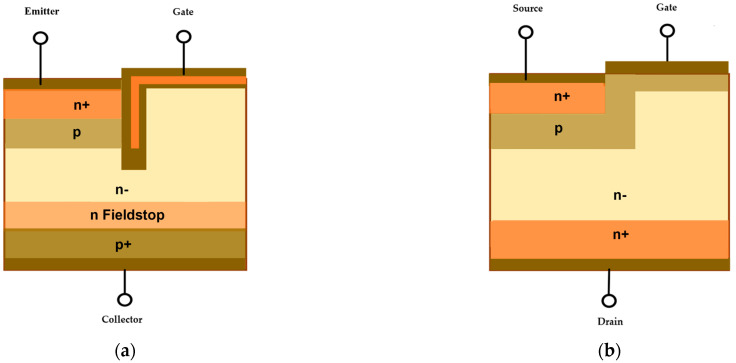
Schematic diagrams for; (**a**) Si IGBT and (**b**) SiC MOSFET.

**Figure 4 micromachines-14-01636-f004:**
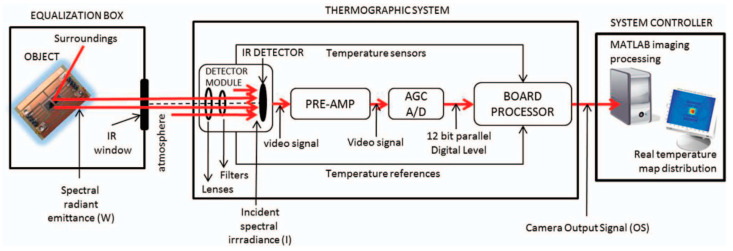
Components of a typical IRC: equalization box, thermographic system, and the system controller Reprinted with permission from Ref. [50], 2023, AIP.

**Figure 5 micromachines-14-01636-f005:**
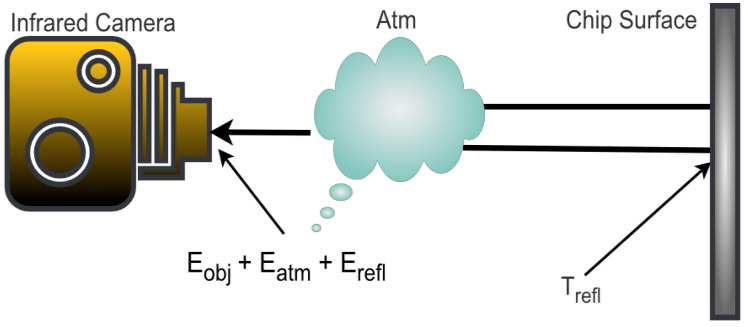
Representation of equivalent radiation captured by IRC sensing technique.

**Figure 6 micromachines-14-01636-f006:**
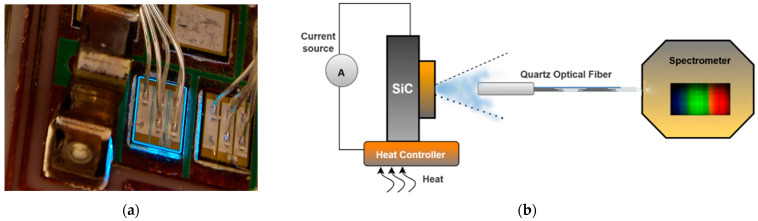
(**a**) EL of a SiC MOSFET body diode. Reprinted with permission from Ref. [83]. 2023, IEEE. (**b**) Generic schematic of TSOP sensing technique with SiC MOSFET module.

**Figure 8 micromachines-14-01636-f008:**
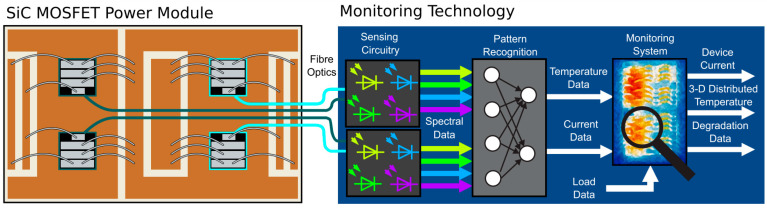
Schematic representation of simultaneous junction temperature and current sensing approach for paralleled SiC MOSFET module Reprinted with permission from Ref. [83], 2023, IEEE.

**Figure 9 micromachines-14-01636-f009:**
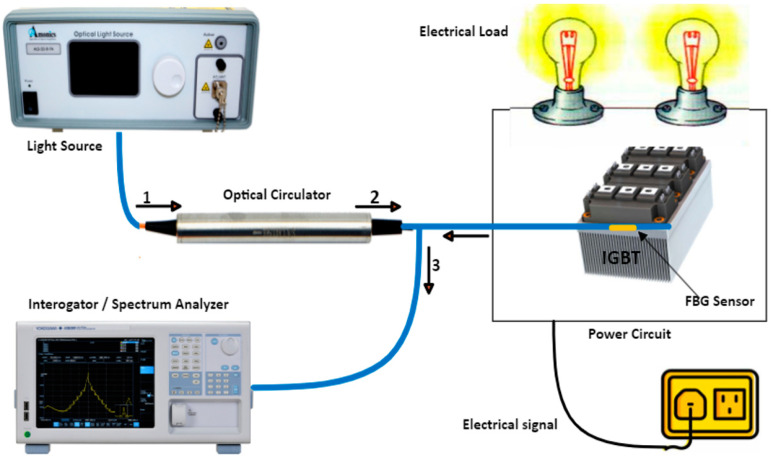
Experimental setup for the FBG-based sensing technique for temperature measurement.

**Figure 10 micromachines-14-01636-f010:**
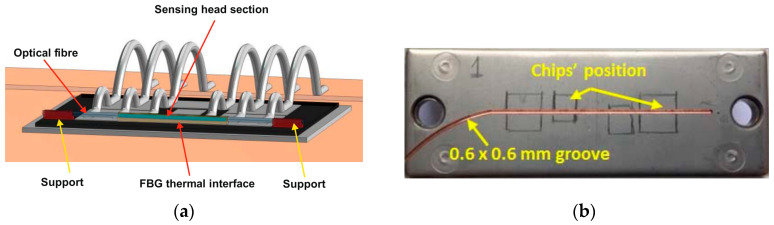
(**a**) Direct on-chip fiber placement for junction temperature sensing. Reprinted with permission from Ref. [114], 2023, IEEE. (**b**) Optical fiber installation on a grooved Si IGBT module. Reprinted with permission from Ref. [115], 2023, IEEE.

**Figure 11 micromachines-14-01636-f011:**
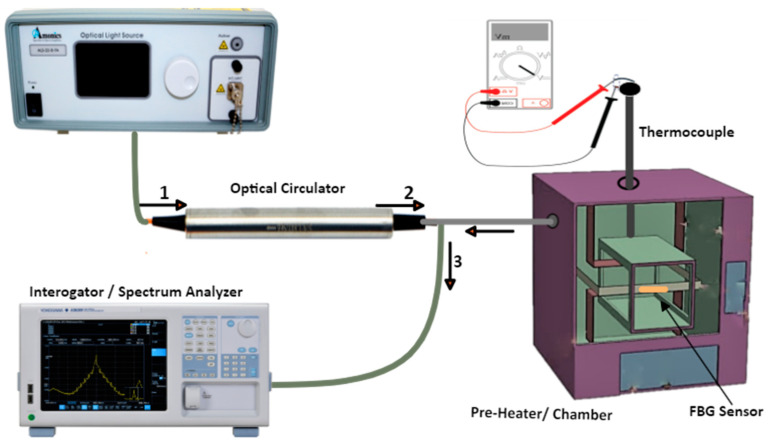
Calibration setup for an FBG-based sensing technique.

**Figure 12 micromachines-14-01636-f012:**
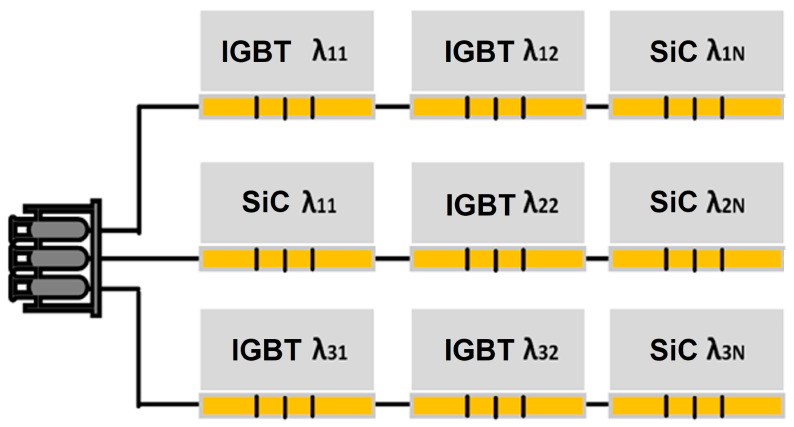
Schematic of distributed temperature sensing using FBG-based technique.

**Table 1 micromachines-14-01636-t001:** Comparison of junction temperature sensing techniques for power switching semiconductor devices.

Characteristics	Optical	Electrical	Physical
Measurand	Light Signal	Electrical Parameters	Physical Quantities
EMI Immunity	Yes	No	Yes
Physical Contact	Maybe	Yes	Yes
Invasive	No	Yes	Yes
Accuracy	High	Moderate	High
Complexity	No	Maybe	No
In-situ Sensing	Yes	Yes	No
Response Time	Moderate	Fast	Poor
Cost	Low	High	Low

**Table 3 micromachines-14-01636-t003:** Summary of key benefits and underlying challenges of IRC sensing technique.

Key Benefits	Challenges
Temperature changes can be easily sensed at some distance.	Junction temperature sensing requires the removal of the power semiconductor package for radiation detection.
Suitable for offline thermal extraction of power semiconductor devices.	To obtain an accurate measurement, even emissivity of the surface is required across the region.
Modern IRC sensors have high spatial resolution and, thus, more accurate results.	Even though thermal imaging can be detected at a distance, it requires clear sight of the object for accurate results.
IRC renders attractive temperature mapping with temperature range bar.	Thermal IRC sensors are external to the system and difficult to embed in the power switches.

**Table 4 micromachines-14-01636-t004:** Advancements in TSOP sensing technique for junction temperature measurement of semiconductor devices.

Year	Device	I (A)	λ (nm)	S	Err	ΔT	Contribution	Ref.
2022	SiC M.	3–18	390/510	-	±3 °C	50–130 °C	Decoupled intensity peaks.	[107]
2021	SiC M.	14	360/420	-	1.5 °C	45–105 °C	Impact of EL biasing and BTI	[95]
2021	SiC M.	2–12	-	-	2.18 K	10–120 °C	Simultaneous device Temperature and current EL extraction	[106]
2020	SiC M.	20, 80	380/480	-	-	30–150 °C	Peak spectral extraction	[108]
2020	SiC M.	5, 20	390/510	-	1.2 °C	10–90 °C	Multiple photodetection	[96]
2020	SiC M.	4–11	380	0.10 mV/°C	3.9 °C	40–100 °C	Embeddable detector	[109]
2019	SiC M.	5	-	3.2 mV/K	-	30–150 °C	Intensity-temperature model	[105]
2019	SiC M.	20, 25	383/485	1.53 V/°C	±5 °C	95–135 °C	Temperature transient	[104]
2018	SiC M.	5	500	-	-	30–130 °C	Photomultiplier for detection	[87]
2018	SiC M.	5–12	387	-	-	25–100 °C	Simple photodetector	[110]
2015	SiC M.	16	420/390			290–800 K	EL—temperature profile	[98]
2013	SiC M.	9	480/390			290–800 K	Diffusion current in SiC structure	[97]

SiC M: Silicon Carbide MOSFET, λ: spectral peaks (blue-green/UV band), I: input current, ΔT: temperature range, Err: measurement error.

**Table 5 micromachines-14-01636-t005:** Summary of key benefits and underlying challenges of TSOP sensing technique.

Key Benefits	Challenges
Online junction extraction.	Low accuracy since EL effect cannot be fully extracted.
Low-cost photodetection circuits can be employed for detection.	High-resolution spectrometer is required for accurate thermal estimation.
Noninvasive to the device operation.	It may become complex, especially when multipoint sensing is required.

**Table 6 micromachines-14-01636-t006:** Advancements in FBG sensing technique for junction temperature measurement of switching semiconductor devices.

Year	Device	λ (nm)	FS (Hz)	Sensor Casing	Sensitivity (pm/°C)	L (mm)	Err	ΔT (°C)	Key Features	Ref.
2022	IGBT-PI	1540	1 M	Teflon tube	11.7	3	0.03 °C	160	Sensor head interface	[68]
2022	MOSFET	1510–1590	1 k		14, 21	15	±0.5 K	-	Thermal model for duty cycles	[112]
2022	IGBT	1550	2 k	-	13	1–5	±0.1 °C	40–67	Head length influence	[114]
2022	IGBT-PI	1550	100	Tube Metal Plate	10.2, 10.4, 14.7	-	-	175	FBG housing effect	[47]
2021	IGBT	1545	-	Tube	-	15	-	-	Transient temperature monitoring	[111]
2020	IGBT	1550	2.5 k	Tube	10.9	5	-	25–95	Steady and dynamic response	[115]
2017	IGBT	1530	100	Teflon tube	-	5	0.48 °C	40–180	Influence of vibration	[116]
2012	IGBT	-	2 k	-	-	-	±0.02 °C	<150	RC model	[113]
2011	IGBT-SI	1550		Metal strip	14			26–90	Established FBG sensitivity threshold	[117]
2011	IGBT-SI	1555	-	Metal sheet	-	-	-	0–80	Interface conductivity	[118]
2010	-	1549	10		11.4	5		30–900	High-temperature sensing	[119]
2010	-	1530		Steel Tube	21 28	-	-	-	FBG sensitivity enhancement	[120]

λ: wavelength, FS: sampling rate, L: grating length, Err: measurement error, ΔT: temperature range, PI: PWM inverter, SI: solar inverter.

**Table 7 micromachines-14-01636-t007:** Summary of key benefits and underlying challenges of FBG sensing technique.

Key Benefits	Challenges
FBG has high resolution and accuracy up to 0.1 °C.	FBG must be placed closer to the wafer to obtain accurate results.
FBG sensors have good stability and a large temperature measurement range.	Accuracy is dependent on interrogator’s resolution.
FBG can be stuck to the device even when the chip is under its operation condition without disturbance.	Thermal adhesive is vital to enhance FBG-IGBT interface for accurate measurement.

**Table 8 micromachines-14-01636-t008:** Comparison between IRC, TSOP, and FBG sensing techniques.

Features	IRC	TSOP	FBG
Operation principle	Infrared radiation	Photon extraction	Wavelength shift
Physical contact	No	Yes	Yes
Accuracy	High	Moderate	High
Signal processing	Moderate	Complex	Simple
Noise	High	Moderate	Low
Temperature range	−40 to 250 °C	25 to 140 °C	−50 to 956 °C
Calibration	One-time	System-dependent	One-time

## Data Availability

Not applicable, as no datasets were generated or analyzed for this research work.

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
