# Peer review of "Junction Temperature Optical Sensing Techniques for Power Switching Semiconductors: A Review"

_micromachines, 2023, doi:10.3390/mi14081636_

Round 1

Reviewer 1 Report

This article reviews three methods for transistor junction temperature sensing and explains the advantages and disadvantages of the different methods. Although rich in content, the technical overview is relatively superficial. Moreover, the writing quality of the entire manuscript was poor, with serious confusion in the legends and table notes, and inconsistent formats of tables and formulas, which caused serious inconvenience to readers. In addition, there are some grammatical inadequacies. Authors should realize that the role of a reviewer is to review your article for technical innovation, not to provide high-volume editing services or proofread drafts. In summary, the manuscript needs major improvements in writing quality. Some comments are summarized below:

1. As a review article, in addition to presenting some work to readers in the form of table summary. The author should present the key pictures of some representative works to readers, so that readers can intuitively grasp the core ideas of the work.

2. A total of 6 addresses appear in the place where the author's address is associated, but in the author's information, there is no author-associated address 5.

3. In the abstract, the author explained the following two terms "thermal sensitive optical parameter (TSOP), and fiber Bragg grating (FBG) temperature sensing" and used abbreviations to refer to them. However, on the third page, the explanation of the same term is repeated again, which is easy to cause misunderstandings for readers. There are many problems like this, such as the first occurrence of "BTJs" on the second page, and the author does not explain the term.

4. Some tables in the text use three-line tables, and some tables use second-line tables. Some tables are bolded and some are not. The formula on page 7 of the text is bolded, while the rest of the formula is not bolded. Please check this type of question carefully against the formatting requirements of the Micromachines journal.

5. In the text, after Figure 5, it is named again from Figure 1, and the expression in the text is seriously inconsistent with the legend. The table following Table 3 in the text is named Table 1. There is serious confusion in legends and table notes.

6. The note in Table 5 is not an explanation of the contents of the Table.

7. The author should format the content of the article reasonably, and there should not be a large blank like page 17. Authors should carefully check the format of the manuscript and make revisions according to the journal's format.

Authors should carefully check the grammar and presentation of the full text.

Author Response

We would like to express our sincere gratitude to the reviewers for their valuable comments and suggestions, which have significantly improved the quality and clarity of our work. We have incorporated the revisions suggested by the reviewers. The revised version now provides well-arranged figures and tables, offering improved clarity and interpretation of the results. In addition, we have taken great care to ensure that the revised manuscript adheres to the journal’s format. The corrections are highlighted in yellow in the revised version. We believe that these changes have strengthened the manuscript, making it suitable for publication.

  1. As a review article, in addition to presenting some work to readers in the form of table summary. The author should present the key pictures of some representative works to readers, so that readers can intuitively grasp the core ideas of the work.

Response: Thanks to the worthy reviewer for the suggestion. The table summary has been refined and revised to make it more understandable to readers.

  1. A total of 6 addresses appear in the place where the author's address is associated, but in the author's information, there is no author-associated address 5.

Response: By mistake, the author’s address 1 was duplicated for address 5. The duplicate addresses have been resolved in the revised manuscript.

  1. In the abstract, the author explained the following two terms "thermal sensitive optical parameter (TSOP), and fiber Bragg grating (FBG) temperature sensing" and used abbreviations to refer to them. However, on the third page, the explanation of the same term is repeated again, which is easy to cause misunderstandings for readers. There are many problems like this, such as the first occurrence of "BTJs" on the second page, and the author does not explain the term. –

Response: Repeated abbreviations have been removed, and omitted ones are now explained in the revised version to avoid confusion.

  1. Some tables in the text use three-line tables, and some tables use second-line tables. Some tables are bolded and some are not. The formula on page 7 of the text is bolded, while the rest of the formula is not bolded. Please check this type of question carefully against the formatting requirements of the Micromachines journal.

Response: These errors have been taken care of in the revised version, and all tables are now in a uniform format.

  1. In the text, after Figure 5, it is named again from Figure 1, and the expression in the text is seriously inconsistent with the legend. The table following Table 3 in the text is named Table 1. There is serious confusion in legends and table notes.

Response: The Figures are now numbered properly and discussed in the text accordingly. Also, the tables are now numbered accordingly in the revised version. The table notes are also updated for the clarity of the readers.

  1. The note in Table 5 is not an explanation of the contents of the Table.

Response: Thanks again for pointing out this issue. The caption of the table was messed up during the formatting process and is now corrected in the revised manuscript.  

  1. The author should format the content of the article reasonably, and there should not be a large blank like page 17. Authors should carefully check the format of the manuscript and make revisions according to the journal's format.

Response: We thank the reviewer for pointing out the format issue. We have now tried our best to make sure that there are no large blank spaces in the revised manuscript.

We have also fixed a few typos found in the revised manuscript.

Reviewer 2 Report

The authors presented an in-depth overview of junction temperature measurement optical techniques. The paper includes a significant number of references (120), which is required and sufficient for review articles. The authors do not present any novel method or improvement of known one made by them, but it is not necessary for such kind of article. In the end, the authors show some ways of development of the junction temperature optical sensing techniques which are interesting.

Review articles are needed to systematize knowledge on a given topic. In my opinion, this article is a good example of a review article and can be accepted with a few technical corrections as follows:

1. There are some Chinese letters in lines 329 and 496 - they should be updated

2. It would be nice if the author could add something about the Lukas et al [94] method mentioned in line 385

3. In some cases the Celsius degree has wrong format e.g. in line 478 or 564, etc.

4. The units format is not uniform e.g. sometimes is 3oC, or 4um sometimes 3 oC, 4 um

5. On page 19 there is once again Figure 2 - it should be Figure 7, and on page 18 there should be Figure 8

Author Response

We would like to express our sincere gratitude to the reviewers for their valuable comments and suggestions, which have significantly improved the quality and clarity of our work. We have incorporated the revisions suggested by the reviewers. The revised version now provides well-arranged figures and tables, offering improved clarity and interpretation of the results. In addition, we have taken great care to ensure that the revised manuscript adheres to the journal’s format. The corrections are highlighted in yellow in the revised version. We believe that these changes have strengthened the manuscript, making it suitable for publication.

1. There are some Chinese letters in lines 329 and 496 - they should be updated.

Response: These errors occurred while changing the Word document to pdf. Now these errors are resolved in the revised manuscript.

2. It would be nice if the author could add something about the Lukas et al [94] method mentioned in line 385.

Response: Thanks for highlighting this issue. The method proposed by Luas et al [94] is now explained in the revised manuscript.

3. In some cases the Celsius degree has wrong format e.g. in line 478 or 564, etc.

Response: These errors have been taken care of in the revised version.

4. The units format is not uniform e.g. sometimes is 3oC, or 4um sometimes 3 oC, 4 um

Response: The format for different units is now unified in the revised manuscript.

5. On page 19 there is once again Figure 2 - it should be Figure 7, and on page 18 there should be Figure

Response: Thanks for pointing out this error. We have now numbered the Figures accordingly in the revised version.

Round 2

Reviewer 1 Report

The author's efforts to respond to the reviewer's questions are appreciated, but reviewers have concerns about the quality of the full text:

1. As a review article, the innovation of the article and the depth of analysis of the three types of junction temperature sensors are not enough.

2. As a review article, there are almost no picture citations to the references, which is less readable for readers. Only through a few statistical tables can not effectively understand the core ideas of the literature.

The author's English expression is commendable.

Round 3

Reviewer 1 Report

I strongly recommend the author to improve the quality of the pictures in the manuscript. English expression can be accepted.

Author Response

We are writing to provide an updated version of our manuscript titled "Junction Temperature Optical Sensing Techniques for Power Switching Semiconductors: A Review" (Manuscript ID: micromachines-2495474, which incorporates the revisions suggested by our respected reviewers. We would like to express our sincere gratitude to the reviewers’ for their valuable comments and suggestions, which have significantly improved the quality and clarity of our work.

  1. I strongly recommend the author to improve the quality of the pictures in the manuscript.

Response: Based on the reviewer's recommendations, we have improved the quality of all the pictures in the revised manuscript.

We also had the text read by a native English speaker to improve the manuscript's flow. Furthermore, we have taken great care to ensure that the revised article follows the format of the journal. We believe that these changes have strengthened the manuscript, making it suitable for publication. We appreciate the opportunity given to revise our manuscript and are grateful for the valuable feedback provided by the reviewers. We look forward to hearing from you soon.

Yours sincerely,

Prof. Khurram Karim Qureshi